# Cerebrospinal Fluid Analysis in Rheumatological Diseases with Neuropsychiatric Complications and Manifestations: A Narrative Review

**DOI:** 10.3390/diagnostics14030242

**Published:** 2024-01-23

**Authors:** Massimiliano Castellazzi, Raffaella Candeloro, Maura Pugliatti, Marcello Govoni, Ettore Silvagni, Alessandra Bortoluzzi

**Affiliations:** 1Department of Neurosciences and Rehabilitation, University of Ferrara, 44121 Ferrara, Italy; raffaella.candeloro@unife.it (R.C.); maura.pugliatti@unife.it (M.P.); 2Department of Medical Sciences, University of Ferrara, 44121 Ferrara, Italy; marcello.govoni@unife.it (M.G.); ettore.silvagni@unife.it (E.S.); alessandra.bortoluzzi@unife.it (A.B.)

**Keywords:** cerebrospinal fluid, rheumatological diseases, neuropsychiatric complications

## Abstract

The analysis of cerebrospinal fluid (CSF) remains a valuable diagnostic tool in the evaluation of inflammatory and infectious conditions involving the brain, spinal cord, and meninges. Since many rheumatic inflammatory diseases can involve the central and peripheral nervous system, the aims of this narrative review were to summarize the latest evidence on the use of CSF analysis in the field of neuropsychiatric manifestations of rheumatic diseases. Routine CSF parameters were taken into consideration for this review: appearance; total protein and cellular content (pleocytosis); lactate and/or glucose; CSF/serum albumin quotient; intrathecal synthesis of IgG. Data regarding the role of CSF analysis in the clinical management of neuropsychiatric systemic lupus erythematosus, primary Sjogren’s syndrome, rheumatoid arthritis, and Behçet’s syndrome are presented. Although no disease-specific picture has been identified, CSF analysis remains a useful diagnostic tool to confirm the presence of a neuro-inflammatory state or, conversely, to exclude the concomitant presence of other inflammatory/infectious diseases affecting the CNS in the context of systemic rheumatologic conditions.

## 1. Introduction

Rheumatic inflammatory diseases encompass a wide array of systemic conditions characterized by multisystem involvement, often driven by autoimmunity and inflammation. They include, among others, connective tissue diseases (CTDs), chronic inflammatory arthritis, and systemic vasculitides (Figure 1) [1].

One significant facet of systemic autoimmune rheumatic diseases is their remarkable proclivity for affecting the nervous system, which can manifest at any stage of the disease, presenting significant diagnostic and therapeutic challenges for clinicians [2]. Neurological complications can present across a broad spectrum, involving both the central (CNS) and peripheral nervous system (PNS), ranging from acute inflammatory neuropathies characterized by rapid-onset nerve inflammation to chronic and subtle cognitive impairments, each presenting a unique spectrum of complexities [3]. These manifestations are driven by diverse inflammatory attacks on neural tissues. Contributing mechanisms include immune complex deposition, orchestration of pro-inflammatory cytokine cascades, and the emergence of aberrant autoimmune responses targeting the neural structures [4].

The analysis of cerebrospinal fluid (CSF) remains a valuable diagnostic tool in the evaluation of inflammatory and infectious conditions involving the brain, spinal cord, and meninges, as well as computed tomography (CT)-negative subarachnoid hemorrhage and leptomeningeal metastases [5]. This has led to considering the analysis of CSF as a sort of liquid biopsy of the brain fluids [6].

## 2. Cerebrospinal Fluid and the Blood-Cerebrospinal Fluid Barrier

CSF is a watery and colorless biological fluid that permeates the CNS. About 70% of the CSF is secreted by the choroid plexuses, special tissues rich in blood vessels that line the cerebral ventricles [7].

Two very important functions are attributed to CSF: (i) mechanical: the CSF has the task of protecting the brain and spinal cord from mechanical stresses linked to variations in movement and position of the head in space; (ii) biological: CSF represents the organ of cerebral homeostasis as it has been observed that its physicochemical characteristics present stability in physiological conditions while they may be altered in various pathologies of the CNS and PNS [8].

CNS is protected from harmful substances contained in the blood mainly by two structures: (i) the blood-brain barrier (BBB), located between the vascular system and the brain parenchyma, and (ii) the blood-CSF barrier at the level of the choroid plexuses in the cerebral ventricles [9,10,11].

The choroid plexuses are epithelial-endothelial complexes comprising a highly vascularized stroma surrounded by connective tissue, and epithelial cells joined together by tight junctions [12].

The blood-CSF barrier, with its different anatomical structures, actively regulates the diffusion and filtration of macromolecules from blood to CSF. The CSF protein content is determined and maintained by the integrity of the blood-CSF barrier, together with the CSF flow rate itself [13]. In preterm and full-term newborns, protein concentrations in the CSF are high but tend to gradually decrease during the first year of the child’s life, then remaining at low levels throughout childhood. In adults, protein concentrations in the CSF increase with age [14,15]. The albumin quotient (QAlb), determined by the albumin CSF/serum ratio, is used to evaluate the integrity of the blood-CSF barrier [16]. QAlb is not influenced by intrathecal protein synthesis, as CSF albumin is entirely plasma-derived and is an integral part of the mathematical formulas for the quantification of intrathecal synthesis of immunoglobulins. The QAlb is a value independent of the method used for its determination. The CSF and serum albumin concentrations are mainly determined with nephelometry or turbidimetry, allowing the use of the same reference values in different laboratories [17,18].

CSF protein concentration and QAlb increase along the circulation of the CSF within the neuraxis, presenting lower concentrations at the ventricular level and higher concentrations in the lumbar sac, creating what is effectively defined as a “rostro-caudal gradient” of protein [19].

## 3. Cerebrospinal Fluid Analysis

CSF is withdrawn relatively easily through a minimally invasive procedure called lumbar puncture [5].

CSF composition is similar to a plasma ultrafiltrate, although with important differences as it has a reduced protein content and very few nuclear cells, mainly lymphocytes and monocytes [8].

A standard CSF analysis should include (i) the evaluation of CSF appearance (e.g., to exclude the presence of erythrocytes), the determination (ii) of total protein and (ii) cellular content and (iii) the concentration of lactate and/or glucose [5,20,21]. Furthermore, the functionality of the blood-CSF barrier should be assessed by calculating the QAlb, and the presence of intrathecal synthesis of IgG should be verified through the use of mathematical indexes or, better, by searching oligoclonal IgG bands in CSF and serum with the use of isoelectrofosing on agarose gel followed by IgG-specific immunofixation [5,6,16,20]. In the presence of a strong suspicion of bacterial or viral infection, bacteriological (Gram stain), culture, and gene amplification (PCR) investigations should be performed [5,20].

## 4. Objectives

The aim of this narrative review is to offer an updated scientific, in-depth analysis, with a strong clinical orientation, on the use of CSF analysis in the field of neuropsychiatric complications and manifestations that can occur during systemic rheumatic inflammatory diseases.

## 5. Methods

The bibliographic search was carried out in September 2023 using the PUBMED search engine: https://pubmed.ncbi.nlm.nih.gov (accessed on 17 January 2024). The keywords “cerebrospinal fluid” and “cerebrospinal fluid analysis” were combined using the Boolean operator “AND” with the names of systemic rheumatic inflammatory pathologies mainly presenting neurological complications or manifestations. The English language and the human species were also used as filters.

Only routine CSF parameters were taken into consideration for this review: (i) appearance; (ii) total protein and (iii) cellular content (pleocytosis); (iv) lactate and/or glucose; (v) CSF/serum albumin quotient (QAlb); (iv) intrathecal synthesis of IgG (Table 1).

## 6. Results: CSF Analysis in Systemic Rheumatic Inflammatory Diseases

Our bibliographic search produced results for the following pathologies: systemic lupus erythematosus, Sjögren’s syndrome, rheumatoid arthritis, and Behçet’s syndrome.

### 6.1. Neuropsychiatric Systemic Lupus Erythematosus

Neuropsychiatric (NP) involvement in systemic lupus erythematosus (SLE) is a potentially severe manifestation of the disease, encompassing several different clinical syndromes affecting both the CNS and PNS [3,22,23,24,25]. NP manifestations can be focal or diffuse, acute or chronic, single or multiple, occurring at each stage of the disease or even preceding it, in association with non-neurological manifestations of SLE, or eventually as an isolated event. In 1999, the American College of Rheumatology (ACR) published a consensus document enlisting 19 NP syndromes (12 CNS and 7 PNS) accredited as potentially related to SLE (Table 2) [26].

A definition was also provided for NPSLE: “Neuropsychiatric lupus erythematosus includes the neurologic syndromes of the central, peripheral, and autonomic nervous system and the psychiatric syndromes observed in patients with SLE in which other causes have been excluded”. In 2010, a set of recommendations was provided by the European Alliance of Associations for Rheumatology (EULAR), addressing diagnosis, prevention and treatment [27]. Given the overall low specificity of NP events for SLE, it is mandatory in clinical practice to evaluate how plausible the attribution of a new NP event to SLE, excluding other more proper diagnoses or confounding factors. After the first attribution models proposed by Hanly et al. [28,29], the Study Group on NPSLE of the Italian Society for Rheumatology (SIR) provided a new attribution algorithm for NP events [30,31]. Overall, one-third of the NP events occurring in SLE patients can reasonably be attributed to SLE using the algorithm.

Furthermore, in order to maximize a correct attribution of NP events occurring in SLE patients, a multidisciplinary approach and a regular tight reassessment are of utmost importance [32]. According to the last updated EULAR therapeutic recommendations for SLE [33,34], it is suggested to tailor the treatment to the presumed underlying pathophysiological mechanism (e.g., inflammatory or embolic/thrombotic/ischemic) [35], with glucocorticoids (GCs) and/or immunosuppressants in case of inflammatory background and anticoagulant/antithrombotic therapy in case of thrombotic phenotypes. Secondary prevention of focal NP disease attributed to aPL antibodies requires lifelong anticoagulation [36]. Moreover, a combination of immunosuppressants and anticoagulant/antithrombotic therapy should be considered when both mechanisms are thought to coexist. The positive effect of hydroxychloroquine has been demonstrated in several studies, while specific symptomatic therapy is indicated according to the type of manifestation (e.g., antipsychotics, anti-depressants, anti-convulsants) [22]. The lack of properly validated outcome measures to be assessed in RCTs remains one of the main limitations to testing newer drugs in this challenging disease [37].

#### 6.1.1. Pathogenesis of Neuropsychiatric Systemic Lupus Erythematosus

Under a genetic background [38], several pathogenetic mechanisms are recognized as the basis of NPSLE, but two different main pathways are suggested, e.g., ischemic injury and inflammation-mediated damage, which are probably intimately connected. Regarding the former, intra-cranial vessel thrombosis, mediated by antiphospholipid antibodies (aPLs), immune complexes, and complement activation, is responsible for large- or small-size infarcts and diffuse small vessel vasculopathy, while cerebral vasculitis is thought to account for only a minor part of the disease burden. This was confirmed by post-mortem studies [39,40], as well as in animal models of NPSLE [41], suggesting an inflammatory complement-dependent component of the thrombotic injury [39]. With respect to inflammation, mechanical or functional BBB alterations [42,43], mainly dependent on systemic inflammatory mediators release or autoantibodies-mediated damage, are responsible for the increase in permeability of the BBB, with serum cytokines (e.g., type I interferons) passing into the CNS, and autoantibodies-dependent neuronal damage or dysfunction (anti-ribosomal P, anti-NR2, and others). This, coupled with microglial activation and intrathecal cytokines production, further corroborates damage inside the neuroaxis [44,45,46].

#### 6.1.2. Cerebrospinal Fluid Analysis in Neuropsychiatric Systemic Lupus Erythematosus

The presence of an inflammatory condition within the CNS has been highlighted through CSF analysis in patients suffering from NPLSE [47]. An altered function of the blood-CSF barrier, demonstrated by high QAlb values, has been found in 30–60% of patients [47,48,49]. The presence of intrathecal synthesis of antibodies has also been demonstrated with quantitative methods and through the presence of oligoclonal bands (OCBs) IgG restricted to the CSF in 64 and 28% of cases, respectively [48,50]. As with some neurological disorders such as multiple sclerosis, even in NPSLE patients, the presence of OCB IgG was not necessarily associated with barrier damage [50]. Of note, barrier dysfunction was found to be an independent risk factor for the development of corticosteroid-induced psychiatric disorders [49]. In patients presenting demyelinating syndrome in SLE, in particular neuromyelitis optica spectrum disorders, CSF analysis showed pleocytosis and hyperproteinorrachia in almost 80% of cases. In contrast, oligoclonal bands were found in less than a quarter of patients [51]. Moreover, CSF analysis is useful in inflammatory demyelinating polyradiculoneuropathy in the context of SLE. In line with the EULAR recommendations, depending upon the type of neuropsychiatric manifestation, lumbar puncture and CSF analysis are suggested, primarily to exclude CNS infections [27].

### 6.2. Sjogren’s Syndrome

In primary Sjogren’s syndrome (SS), neurological manifestations can occur in a significant proportion of patients, with the PNS being the most frequent site of involvement compared to the CNS [52,53]. With regard to the latter, the real burden of CNS involvement in SS remains elusive, with the prevalence ranging from 0.3 to 9.7% [54,55]. In SS, neurological abnormalities may be subtle, often characterized by insidious onset, with a variable course (intermittent/remittent or slowly progressive). CNS manifestations in SS can be classified into diffuse and focal (e.g., stroke-like) or multifocal (e.g., MS-like) with the involvement of both the brain and the spinal cord (Table 3).

Epilepsy, psychosis, and movement disorders are rare, while sleep disturbances, mild cognitive deficits, and fatigue are more common [56]. Regarding the cognitive manifestations, they are frequently the first clinical manifestation, sometimes preceding the diagnosis of SS. They are tendentially mild (the so-called “*brain fog*”). A frontal-subcortical pattern of memory alteration, attention, and executive disorders has been described [57]. In milder cognitive deficits, the attribution to the underlying SS is quite difficult; in addition, it is well-acknowledged that cognitive impairment is strictly related to pain, and SS patients often exhibit a “fibromyalgic profile or even a true concomitant fibromyalgia” [58].

PNS involvement in SS, instead, encompasses a wide spectrum of peripheral neuropathies. In the most representative cohorts, the estimated frequency ranges between 1.8 and 17% [59,60], and PNS involvement may be the presenting feature in around 25% of cases [61,62]. Distal axonal sensory polyneuropathy and sensorimotor polyneuropathy are the most common; other peripheral neuropathies include small fibers neuropathy (SFN), multiple mononeuropathy, sensory ganglionopathy, cranial nerve neuropathies, chronic inflammatory demyelinating polyneuropathy (CIDP) and motor neuron diseases [53,55]. Dorsal root ganglionitis may selectively affect small neurons and present with painful dysesthesias in an asymmetric, patchy, non-length-dependent distribution [63].

Robust evidence to guide therapeutic decisions in neurologic involvement in SS patients is lacking [64]. Overall, the use of GCs, antimalarials, immunosuppressive agents, and intravenous immunoglobulins (IVIG) should be restricted to patients with active systemic disease. Treatment of CNS involvement in SS remains largely empirical. In the case of peripheral neuropathies, apart from symptomatic measures to relieve neuropathic pain and dysesthesias, GCs and immunosuppressive agents should be used in severe cases, along with IVIG, rituximab, or plasma exchange (PEX) for refractory cases [65,66].

#### 6.2.1. Pathogenesis of Neurological Involvement in Sjogren’s Syndrome

The pathogenesis of CNS involvement in SS has not been clearly defined yet. Some evidence and few histopathological studies support the hypothesis of an immunologically mediated small vessel vasculopathy and, to a lower extent, of a true small vessel vasculitis [59,67,68]. On the other hand, a combination of ganglion neuronopathy and vasculitis has been postulated as the underlying physiopathological mechanism of most of the PNS manifestations [69].

#### 6.2.2. Cerebrospinal Fluid Analysis in Sjogren’s Syndrome

Patients with SS may show consistent signs of inflammation in the CSF without fitting into a specific pattern. In fact, pleocytosis, increased protein levels, and blood-CSF barrier dysfunction (altered QAlb) were detected in less than a quarter of patients with primary and secondary Sjogren syndrome [70]. Patients with primary Sjögren’s syndrome presented peripheral neuropathy with sensory deficits, peripheral neuropathy with additional paresis, cranial neuropathy, myalgia, and CNS involvement. Patients with secondary Sjögren’s syndrome presented an association with other autoimmune diseases [70]. A quarter of the patients with primary form and half of those with secondary form were also positive when tested for CSF-restricted IgG OCB [70]. In a cohort of subjects suffering from Sjögren’s syndrome and neurologic manifestations, an increased pleocytosis and the presence of oligoclonal bands were also reported in one-third of patients [71].

### 6.3. Neurological Involvement in Rheumatoid Arthritis

Among the extra-articular manifestations of rheumatoid arthritis (RA), the neurological complications cover a wide spectrum of clinical pictures spanning from the consequences of the arthritic involvement of the cervical spine to those directly affecting both CNS and PNS (Table 4) [72].

Focusing on this second group of affections, the concept of neuro-inflammation in RA as part of the systemic disease has only recently emerged [73]. High levels of circulating pro-inflammatory cytokines (TNFa, IL-1b, IL-6) are intrinsic parts of the disease, and, in this scenario, IL-6, given its pleiotropic effects and experimental evidence in preclinical models, seems to play a pivotal role in peripheral and central pain sensitization [74], sleep disturbances, stress, fatigue and mood disorders [73,74]. Depression and anxiety are reported in a high number of RA patients with a wide range of frequency, up to 40–70% for mild and moderate symptoms [75,76]. Cognitive dysfunctions (especially in visual-spatial, verbal fluency, logic, and short memory and planning functions) are more frequently observed in RA than in controls and can already be present in the initial stages of the disease [77]. Contrariwise, pachymeningitis secondary to RA inflammation is a rare entity with nonspecific clinical symptoms usually occurring in the setting of long-standing RA [78,79]. Rheumatoid pachymeningitis is usually dissociated from peripheral joint disease activity, and it can occur even in the absence of systemic disease activity [80]. The main clinical features include focal neurologic signs, cranial nerve dysfunction, altered mental status or cognitive dysfunction, seizures, and headache. Pathological features are characterized by chronic inflammation of the meninges (mononuclear cells, especially plasma cells, with occasional necrosis and multinucleated giant cells), rheumatoid nodules, and vasculitis (involving smaller parenchymal and meningeal arteries) [81]. MRI helps to establish the diagnosis easily without biopsy, which can be indicated only in the case of symptom exacerbation or resistance to immunosuppressant therapy [82]. There are no guidelines for the treatment of rheumatoid pachymeningitis. GCs remain the treatment of choice for initial and long-term treatment. Immunosuppressants (cyclophosphamide, azathioprine, methotrexate) have been used in refractory cases; there are also reports of successful treatment with rituximab [83]. Finally, it is well acknowledged that RA patients have an increased risk of cardiovascular and cerebrovascular disease, along with accelerated atherosclerosis, compared with the general population; therefore, a careful assessment of the cardiovascular risk in patients with RA should be part of daily clinical practice [84,85].

Among the PNS manifestations, instead, apart from entrapment/compressive neuropathies which frequently occur in the natural history of the disease and are not under the lens of this review, non-compressive neuropathies (usually related to drugs, vasculitis, or amyloidosis) should be enlisted, including mononeuritis multiplex (associated with rheumatoid vasculitis), distal symmetric axonal sensory neuropathy and sensorimotor neuropathy, demyelinating peripheral neuropathy (associated with the use of anti-TNF drugs) [86] and, rarely, autoimmune autonomic ganglionopathy [87]. Symptomatic medications and physiotherapy are advised. In severe cases of demyelinating neuropathies, GCs and IVIG can be used [87].

#### Cerebrospinal Fluid Analysis in Rheumatoid Arthritis

In 1951, Lush and colleagues described the results obtained from an accurate chemical estimation of CSF and blood protein levels in 23 cases of rheumatoid arthritis [88]. The results indicated that total protein levels measured in the LCS of rheumatoid arthritis patients in both sexes were similar or above average: (i) the increase in total protein was significant only in female patients, although not markedly; (ii) globulins were increased in both sexes, with a significant increase only in male patients; (iii) albumin had normal values in male patients but was elevated in female patients. The globulin/albumin ratio (G/A) in the LCS was therefore overall high. Total protein levels measured in serum were predominantly normal; however, serum globulin and G/A ratios were high in almost all cases. The correlation between total serum protein and serum globulin was very high, while the correlation between total serum protein and serum albumin tended to be zero. It could be deduced that in these cases, the high levels of serum proteins were due solely to the globulin fraction, probably due to systemic inflammation, while changes in the LCS of total proteins, albumin, and globulins tended to occur simultaneously and in the same direction. This phenomenon can be explained by an increased permeability of the blood-brain barrier in patients with rheumatoid arthritis, and, for this reason, serum proteins penetrate the CNS compartment, producing an increase in CSF total protein levels. It should be noted that at the time of the study, the blood-CSF barrier function was not yet evaluated through the CSF/serum albumin quotient.

In the same period, data from patients with a diagnosis of “rheumatoid spondylitis” were also presented [89]. Data from 50 consecutive cases, 33 of which were found to have spondylitis, while in 17 cases, spondylitis was associated with peripheral rheumatoid arthritis, were included in the study. Data obtained from CSF collection showed that the leukocyte count was normal in all cases, protein level was increased in 42% of cases, and serum proteins were normal in most cases. It was also observed that the presence of peripheral arthritis in people with rheumatoid spondylitis also seemed to improve the possibility of an increase in CSF proteins. With regard to the increase in CSF proteins found in patients with rheumatoid spondylitis, the authors considered that (i) it was not necessary that the spine was involved in order to have an increase in CSF proteins at the lumbar level; therefore, the increase of CSF proteins in the rheumatoid spondylitis was not necessarily due to inflammation of the spinal meningeal tissues; (ii) in the case of rheumatoid spondylitis, or rheumatoid arthritis not associated with spondylitis, the increase in CSF proteins could be due to a greater permeability of the choroid plexuses to plasma proteins; (iii) in addition, spinal involvement could significantly increase the chances that a patient with rheumatoid arthritis might have an increase in the amount of CSF proteins because these could reach the subarachnoid space through the perivascular and perineural spaces due to local inflammation of structures near the spine.

In a recent study performed on patients with rheumatoid meningitis, 130 cases of adults affected by RM with an average age of 62 years, with or without a previous diagnosis of RA, were analyzed, and it was observed that the presence of RA and the duration of the disease were associated with a worse prognosis [79]. The most common clinical manifestations were transient focal neurological signs (64.4%), systemic symptoms (51.3%), episodic headache (50.4%), and neuropsychiatric alterations (47.7%), while joint manifestations were present only in 27.4% of cases. CSF analysis showed increased protein level (76.14%), with pleocytosis (85.19%) and mononuclear predominance (89.19%) [79].

### 6.4. Neurological Involvement in Behçet’s Syndrome

Neurological involvement in Behçet’s syndrome, termed Neuro-Behcet’s Syndrome (NBS), primarily affects the CNS, with PNS abnormalities being rare. NBS manifests in approximately 5–10% of all Behçet’s syndrome cases, but the frequency varies widely due to study design, the definition of neurological involvement, and ethnic and geographic diversity. NBS is almost three times more prevalent in men than women, typically emerging between the ages of 20–40 [90].

Neurological symptoms in Behçet’s syndrome often appear a few years after the onset of the syndrome, with an average time ranging from 3 to 6 years between Behçet’s syndrome diagnosis and the development of NBS. Around 75–80% of NBS cases display CNS involvement, referred to as “parenchymal NBS” or “intra-axial NBS”. In approximately 20%, it presents as cerebral venous sinus thrombosis (CVST), also known as “non-parenchymal NBS”, “vascular NBS”, or “extra-axial NBS”. A mixed pattern is reported in 18–20% of patients [91]. Parenchymal NBS affects the telencephalic/diencephalic junction, brainstem (isolated brain stem atrophy, especially in the absence of cortical atrophy, is highly pathognomonic of BS), and spinal cord; usually, it is associated with flaring of systemic BS [92]. Some patients may present only a single acute attack, with or without residual neurologic deficits, but most will experience recurrences with neurological sequelae, and a small number of patients will undergo secondary chronic progression [93]. Subacute meningoencephalitis accounts for 75% of cases in parenchymal NBS, presenting with various clinical features such as headache, cranial neuropathies, dysarthria, ataxia, hemiparesis, cognitive-behavioral changes, dementia, self-limited or progressive myelopathy with sphincter dysfunction, and, to a lesser extent, extrapyramidal signs and seizures, cerebellar degeneration, isolated optic neuritis, or recurrent peripheral facial paresis [94].

Non-parenchymal NBS involves the main vascular structures of the CNS presenting with CVST and arterial involvement. The nature and severity of clinical manifestations of CVST vary according to the site and extent of venous occlusion. CVST usually evolves gradually. Papilledema and sixth nerve paresis are the most common clinical signs related to increased intracranial pressure [95]. Arterial involvement primarily affects large arteries outside the brain, suggesting the existence of an extra-axial arterial pattern of NBS alongside an intra-axial arterial NBS pattern related to intracranial arteritis and intra-axial small arterial occlusions. While aneurysm formation is common in visceral sites, it is extremely rare in intracranial or extracranial arteries [96]. In the absence of randomized controlled trials (RCTs), managing NBS relies heavily on uncontrolled studies and case reports. Initial treatment for acute parenchymal involvement typically involves high-dose glucocorticoids followed by a gradual tapering regimen, as recommended by EULAR guidelines [97]. Azathioprine (2–3 mg/kg/day) has robust evidence in NBS therapy, while TNF inhibitors, particularly infliximab, are suggested options [98]. Cyclosporine is discouraged due to potential neurotoxicity, but interferon-α, methotrexate, and cyclophosphamide are considered in specific cases [97,99]. Emerging evidence highlights tocilizumab and IL-1 inhibitors as potential alternatives for refractory cases, while cautious use of anticoagulants may be considered post-exclusion of arterial aneurysms [100,101].

#### 6.4.1. Pathogenesis of Neurological Involvement in Behçet’s Syndrome

The etiology of NBS remains incompletely understood, with factors such as viral and bacterial agents, immunological responses, genetic predispositions, and fibrinolytic defects being implicated. HLA-B51, combined with cigarette smoking, has been identified as a risk factor associated with chronic and progressive NBS [102].

The basic pathology in the acute/subacute parenchymal presentation of NBS is perivasculitis, characterized by perivascular infiltration of lymphocytes, macrophages, neutrophils, and rarely eosinophils. These infiltrating cells secrete pro-inflammatory cytokines, including TNF-α, IL-1, IFN-γ, IL-6, and IL-18, which may induce vascular endothelial injury and dysfunction, leading to a thrombotic tendency. In later stages, inflammatory infiltration is less prominent, and axonal loss and gliosis predominate [103]. In the pathogenesis of non-parenchymal NBS, there is evidence of endothelial cell activation, with increased concentrations of serum markers of vascular endothelial cell injury (i.e., Von Willebrand’s factor, tissue plasminogen activator and antithrombin III).

#### 6.4.2. Cerebrospinal Fluid Analysis in Behçet’s Syndrome

In parenchymal NBS, the CSF cell count is increased but usually remains below 200/mm^3^. Pleocytosis is dominant in lymphocytes or, less frequently, in polymorphonuclear cells. However, in this condition, CSF cell counts may still be normal [104]. CSF protein levels are frequently higher than normal, and the albumin quotient may result in increased [50,104]. CSF glucose is usually normal, and this may aid in differentiating acute NBS from infectious meningitis [104]. Finally, CSF-restricted IgG oligoclonal bands can be found in a small percentage of patients (10–20%) [50,104]. On the other hand, all CSF parameters are normal, except for the presence of high levels of intracranial pressure in non-parenchymal NBS associated with cerebral venous sinus thrombosis [104,105]. Data regarding the clinical utility of CSF analysis in other systemic vasculitides are relatively scarce, and neurological manifestations are only marginally addressed in the recently released sets of international recommendations [106,107].

## 7. Discussion and Conclusions

To the best of our knowledge, this narrative review represents the first attempt to pool evidence on the clinical use of CSF analysis in patients with rheumatologic inflammatory disorders with CNS/PNS involvement. The presence of CSF alterations, especially as indices of inflammation at the CNS level, has been highlighted in various rheumatologic conditions, as summarized in Table 5.

In patients suffering from NPSLE, the CSF examination could highlight the presence of markers of inflammation in approximately 60% of cases, as evidenced by the presence of augmented protein levels, damage to the blood-liquor barrier, pleocytosis and the presence of an intrathecal IgG synthesis [47,48,49,50]. In some NPLSE patients, clinical relapse was associated with worsened blood-CSF barrier impairment, increased total CSF protein, and, in one patient, the development of serum oligoclonal IgG. Treatment with steroids or immunosuppressants has been associated with an improvement in blood-brain barrier function and the disappearance of intrathecal synthesis of oligoclonal IgG [50].

In patients with neurological diseases associated with SS, CSF markers of inflammation were observed in approximately 25–50% of patients without a characteristic pattern being found, regardless of the peripheral or central genesis of the neurological defects [70,71].

Although studies on patients suffering from rheumatoid arthritis date back to the 1950s, a period in which the functionality of the blood-CSF barrier was not yet assessed using the QAlb, the authors noted an increase in CSF proteins in the absence of a corresponding increase of serum levels in approximately 40% of patients of both sexes [88,89]. Most likely today, given the use of the CSF/serum albumin ratio, this increase would be assessed as a consequence of an altered barrier function. It should also be highlighted that in patients presenting with rheumatoid meningitis, an increase in CSF protein content and the presence of pleocytosis can be found in 75 and 80% of cases, respectively, confirming a greater involvement of inflammation at the CNS level in these patients [79,108]. Also, the presence of local synthesis of IgG was described in a case report relating to a 62-year-old female patient suffering from rheumatic meningitis [108]. Of note, in that patient, treatment with high-dose steroids and Rituximab resolved clinical symptoms and proved capable of bringing back to normal all the CSF inflammatory parameters such as pleocytosis, protein levels, and intrathecal IgG synthesis [108].

CSF analysis highlights different profiles in parenchymal NBS compared to non-parenchymal NBS. In the first form, in fact, increases in CSF protein content, high levels of QAlb, pleocytosis, and, in some cases, intrathecal synthesis were found, albeit in less than half of the patients. The non-parenchymal form, however, showed no signs of inflammation, but an increase in CSF pressure could be found [50,104,105]. In one patient with NBS, steroid treatment resulted in improved blood-brain barrier function and a reduction in CSF infiltrating white blood cells but no alteration in the CSF oligoclonal profile [50].

The main limitation of this narrative review is undoubtedly the small number of articles found in the literature, which highlights how this field is still partially unexplored. However, taken together, the information found allows us to clearly outline some aspects of the role of neuroinflammation in rheumatological diseases with neuropsychiatric complications.

Although no disease-specific pictures have been identified, CSF analysis can be a useful diagnostic tool to confirm the presence of a neuroinflammatory state or, conversely, to exclude the concomitant presence of other inflammatory and infectious pathologies affecting the CNS. Future research can offer deeper insights into CSF analysis for rheumatic conditions with neuropsychiatric complications, potentially introducing validated biomarkers to enhance sensitivity and specificity in CSF investigations.

## Figures and Tables

**Figure 1 diagnostics-14-00242-f001:**
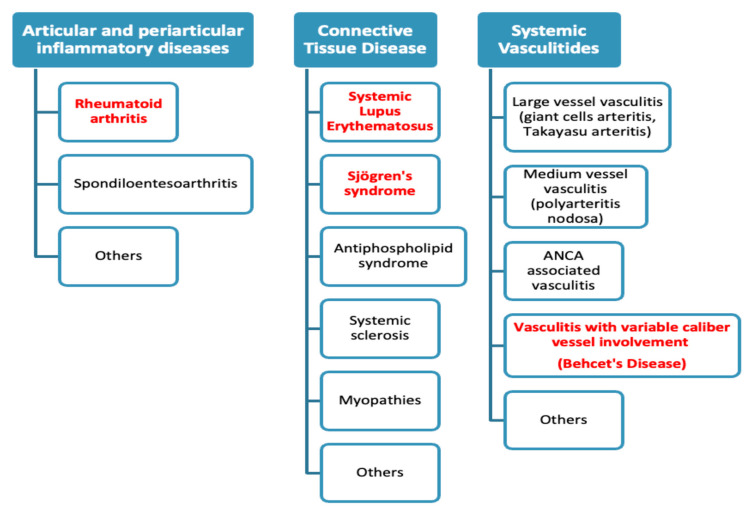
The figure illustrates the classification of articular inflammatory diseases, connective tissue diseases, and systemic vasculitides by the Italian Society of Rheumatology, with the disease discussed in the present review highlighted in red. Available online at: https://www.reumatologia.it/obj/files/AttiCongressi/REUMA_SUPPL_2_2019_ita_LOWRES.pdf (accessed on 17 January 2024).

**Table 1 diagnostics-14-00242-t001:** Parameters of the cerebrospinal fluid (CSF) analysis.

Parameters	Normal/Reference Value	Conditions
Appearance	Clear/colorless	Altered in:
		intracerebral hemorrhage,
		subarachnoid hemorrhage,
		infection (mainly bacterial),
		inflammation
Total protein	450/500 mg/L	Increased in:
		infection (bacterial, viral),
		inflammation,
		metastasis
Cellular content	<5 cells/μL (mainly lymphocytes and monocytes)	Increased in:
		infection (bacterial, viral),
		inflammation,
		metastasis.
Lactate	<1.0–2.9 mmol/L	Increased in:
		infection (bacterial)
Glucose ratio	>0.4–0.5	Reduced in:
		infection (bacterial).
Albumin quotient (QAlb)	<6.5, age 15–40 years,	Increased in:
	<8.0, age 41–60 years	blood-CSF barrier dysfunction,
	<9.0 over 60 years	infection (bacterial, viral)
	Or age/15 + 4	inflammation.
Intrathecal IgG synthesis	Absent	Present in:
		inflammation (acute/chronic),
		infection (mainly viral),
		tumors.

**Table 2 diagnostics-14-00242-t002:** Central and Peripheral Nervous System manifestations according to the 1999 ACR Case Definition for neuropsychiatric systemic lupus erythematosus (NPSLE) [26].

Central NPSLE	Peripheral NPSLE
Aseptic meningitis	Guillain-Barré syndrome
Cerebrovascular disease	Autonomic neuropathy
Demyelinating Syndrome	Mononeuropathy (single/multiplex)
Headache	Myasthenia gravis
Movement disorder	Cranial neuropathy
Myelopathy	Plexopathy
Seizure disorders	Polyneuropathy
Acute confusional state	
Anxiety disorder	
Cognitive dysfunction	
Mood disorder	
Psychosis	

**Table 3 diagnostics-14-00242-t003:** Main manifestations of central and peripheral nervous system involvement in primary Sjögren’s syndrome.

Central Nervous System	Peripheral Nervous System
Focal/multifocal involvement	Distal axonal sensory polyneuropathy (DASP) (the most frequent)
Stroke	Axonal sensory-motor polyneuropathy
NMOSD	Sensory neuronopathy (most typical, but rare): [non-length-dependent sensory neuropathy]
MS-like syndromes	Small fiber neuropathy (SFN)
ALS-like syndrome	Mononeuritis, multiple mononeuropathy
Diffuse abnormalities	Chronic inflammatory demyelinating polyneuropathy (CIDP)
Cognitive dysfunction	Cranial neuropathies
Dementia	Autonomic neuropathy (Adie’s pupils, gastrointestinal abnormal motor activity, bladder dysfunction, orthostatic hypotension, heart arrhythmia, secretomotor dysfunction, anhidrosis)
Psychiatric abnormalities	
Aseptic meningoencephalitis	

ALS: amyotrophic lateral sclerosis; CIDP: chronic inflammatory demyelinating polyneuropathy; MS: multiple sclerosis; NMOSD: neuromyelitis optica spectrum disorders.

**Table 4 diagnostics-14-00242-t004:** Neurological involvement in rheumatoid arthritis (excluded arthritic involvement of the cervical spine and entrapment/compressive neuropathies).

Central Nervous System	Peripheral Nervous System
Sleep disturbances, stress, fatigue	Mononeuritis multiplex
Mood disorders (anxiety, depression)	Distal symmetric axonal sensory neuropathy
Cognitive dysfunction	Distal symmetric axonal sensorimotor neuropathy
Rheumatoid pachymeningitis	Demyelinating peripheral neuropathy
Cerebrovascular disease and accelerated atherosclerosis	Autoimmune autonomic ganglionopathy
Intracranial multiple rheumatoid nodules	
Optic neuritis	
Normal-pressure hydrocephalus	
CNS vasculitis	

**Table 5 diagnostics-14-00242-t005:** Main cerebrospinal fluid alterations found in rheumatological diseases with neuropsychiatric complications.

	Total Proteins	Albumin Quotient	Pleocytosis	Intrathecal IgG Synthesis	Other Parameters
Neuropsychiatric Systemic Lupus Erythematosus	↑	↑ (30–60%)	↑ (30/44%)	Present (30–60%)	
Sjogren’s syndrome	↑ (25–30%)	↑ (<25%)	↑ (25–30%)	Present (25–50%)	
Rheumatoid arthritis	↑ (40% of cases) (75%: in meningitis)		↑ (80%: in meningitis)	Present *	
Neuro-Behçet (parenchymal)	↑ (mildly)	↑ (40%)	↑ (<200/μL)	Present (15–20%)	
Neuro-Behçet (associated with venous sinus thrombosis)					↑ (CSF pressure)

* Case report. ↑, increased.

## Data Availability

No new data were created or analyzed in this study. Data sharing is not applicable to this article.

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
