# Peer review of "Cerebrospinal Fluid Analysis in Rheumatological Diseases with Neuropsychiatric Complications and Manifestations: A Narrative Review"

_diagnostics, 2024, doi:10.3390/diagnostics14030242_

Round 1

Reviewer 1 Report

Comments and Suggestions for Authors

I read with interest the review titled « Cerebrospinal fluid analysis in rheumatological diseases with neuropsychiatric complications and manifestations: a narrative review » .

The review is well-written, clear in its intent, and useful for clinicians and researchers.

A few minor considerations :

In figure1, some diseases are highlighted in red, but no explanation is given in the figure footnote (later, the reader understands that these are these diseases talked in the review). Can you add the explanation in the footnote?

The various tables are very informative.

For cerebrospinal fluid, CSF abbreviation is used. However, please further check the manuscript for missing abbreviations (i.e lines 77 and 79).

My final decision is minor revision.

Reviewer 2 Report

Comments and Suggestions for Authors

This review by Castellazzi et al. provides a valuable summary of the role of cerebrospinal fluid (CSF) analysis in the diagnosis and management of neuropsychiatric complications associated with systemic rheumatic inflammatory diseases (SRIDs). The authors highlight the importance of CSF analysis, given the challenges in the clinical diagnosis of SRIDs with neurological involvement.

The abstract utilizes unnecessary Roman numerals, which could be removed. The review effectively lays the groundwork by introducing relevant background information on CSF, the blood-CSF barrier, and their respective functions. The search methodology employed is clearly outlined and demonstrates precision.

Castellazzi et al. comprehensively summarize and critically discuss existing research on the topic, providing a well-rounded perspective. Their review is further enriched by the inclusion of illustrative figures and tables. However, enhancing the clarity of the tables by adding horizontal lines to separate rows would be beneficial.

While the focus on four specific SRIDs – rheumatoid arthritis, systemic lupus erythematosus, Behcet disease, and Sjögren's syndrome – is appreciated, the rationale for this selection could be further elaborated upon in the review. Please write few sentences that justify your selections.

Overall, this review by Castellazzi et al. is well-written and informative. With the suggested points addressed, it merits publication in this specialized journal.
